# Association between social isolation and depression onset among older adults: a cross-national longitudinal study in England and Japan

Taiji Noguchi [1,2] Masashige Saito [3] Jun Aida [4,5] Noriko Cable [6] Taishi Tsuji [7,8] Shihoko Koyama [9] Takaaki Ikeda [10,11] Ken Osaka [11] Katsunori Kondo [8,12]

For numbered affiliations see end of article.

**Correspondence to**
Mr Taiji Noguchi;
noguchi.taiji0415@gmail.com

## ABSTRACT

**Objective** Social isolation is a risk factor for depression in older age. However, little is known regarding whether its impact varies depending on country-specific cultural contexts regarding social relationships. The present study examined the association of social isolation with depression onset among older adults in England, which has taken advanced measures against social isolation, and Japan, a super-aged society with a rapidly increasing number of socially isolated people.

**Design** Prospective longitudinal study.

**Setting** We used data from two ongoing studies: the English Longitudinal Study of Ageing (ELSA) and the Japan Gerontological Evaluation Study (JAGES).

**Participants** Older adults aged ≥65 years without depression at baseline were followed up regarding depression onset for 2 years (2010/2011–2012/2013) for the ELSA and 2.5 years (2010/2011–2013) for the JAGES.

**Primary outcome measure** Depression was assessed with eight items from the Centre for Epidemiologic Studies Depression Scale for the ELSA and Geriatric Depression Scale for the JAGES. Multivariable logistic regression analysis was performed to evaluate social isolation using multiple parameters (marital status; interaction with children, relatives and friends; and social participation).

**Results** The data of 3331 respondents from the ELSA and 33 127 from the JAGES were analysed. Multivariable logistic regression analysis demonstrated that social isolation was significantly associated with depression onset in both countries. In the ELSA, poor interaction with children was marginally associated with depression onset, while in the JAGES, poor interaction with children and no social participation significantly affected depression onset.

**Conclusions** Despite variations in cultural background, social isolation was associated with depression onset in both England and Japan. Addressing social isolation to safeguard older adults' mental health must be globally prioritised.

## INTRODUCTION

With population ageing, there is growing worldwide interest in social issues concerning older adults, including social isolation and the deterioration of physical and mental health. Defined as an objective state in where an individual has few close relationships or limited contact within a community,[1] social isolation is recognised as a social determinant of health with relevance to mortality,[2] cardiovascular diseases,[3] dementia[4] and mental health.[5 6] Social isolation is a major risk factor for mental health problems in older age. Several systematic reviews have demonstrated that social isolation is associated with depressive symptoms,[5 6] which, in turn, are correlated with unhealthy behaviours and reduced access to material resources.[7] Depression, common in later life, is related to adverse health outcomes such as poor quality of life[8] and functional disability.[9] With the high current global burden of depression expected to increase further by 2030,[10] addressing social isolation is an important gerontological issue for protecting mental health among older adults.

**Strengths and limitations of this study**

► This is the first cross-national longitudinal study to examine the association between social isolation and depression onset in England, which has taken advanced measures against social isolation, and Japan, a super-aged society with a rapidly increasing number of socially isolated people.

► This study included a large sample of over 3300 individuals from England and 33 000 individuals from Japan aged 65 years and older.

► A limitation of this study is that we cannot make direct comparisons because of variations in cohort follow-up periods and depression measurement.

► Another limitation is the use of social support for the evaluation of social contact so as to permit the use of the same social isolation assessment scale in both countries.

BMJ

The impacts of social isolation on health may vary by country; this could be the result of differences in the social environments related to social networks within and outside the family. A recent study of older adults in England and Japan showed that social isolation is a common risk factor for mortality in both countries, with a greater impact observed in England; the results are discussed in terms of possible differences between societies that are highly connected and those that are not.[11] In the UK, in recognition of the impact of social isolation on health and economic loss, the position of 'Minister of Loneliness' was established in 2018, and the country is taking a progressive approach to the elimination of social isolation.[12] In contrast, Japan, now a super-aged society (more than 21% of the population aged 65 or above),[13] is experiencing a rapidly increasing trend in the number of never-married persons and weakening community and neighbourhood relations,[14] leading to a rise in the number of socially isolated individuals.[15] In Japan, the proportion of people who rarely or never spend time with those close to them has been reported to be the highest among Organisation for Economic Co-operation and Development countries. In particular, this figure is much higher than in the UK, which has made advances in tackling social isolation (Japan=15.3%, UK=5.0%).[16] Owing to differences in social structures and the contexts surrounding social isolation, the impact of social isolation on depression is expected to vary across countries.

Furthermore, the health effects of social isolation may differ depending on the cultural context of social relationships. In East Asian countries, including Japan, there is a familial norm based on the traditional culture of filial piety,[17] which is often contrasted with individualism in Western countries.[18 19] Based on this cultural background, Japanese social support networks may be kinship centred, which may be narrower than the types of social networks prevalent in other countries.[20] However, there is a lack of consensus on the health effects of social relationships based on these cultural differences. A previous cross-national study showed that among English men, friendship-based social relationships had a significant impact on longevity, whereas among Japanese men, this impact was associated with family-based social relationships.[21] In contrast, a study of older adults in the USA and Japan demonstrated that while relationships with children were associated with a low level of depression only in Japan, the presence of spouses was important in both countries, but more so in the USA.[22] Another comparative study among adults suggested that social contact with friends benefited women's mental health in the UK but not in Japan.[23] Thus, the family-oriented nature of East Asian societies does not automatically imply the health importance of family-based relationships, and the roles of individual components of social isolation (family, friends and others) in the mental health of older adults in each country remain controversial.

As the association between social isolation and depression is often described as bidirectional,[24] longitudinal studies are needed to address temporality. However, previous cross-national comparative studies have employed only cross-sectional designs.[22 23] Therefore, using longitudinal data from both countries, the present study aims to investigate the association of social isolation with depression onset in England, which has taken advanced measures against social isolation, and Japan, a super-aged society with a rapidly increasing number of socially isolated people.

## METHODS
### Sample
This longitudinal study was conducted using data from two ongoing prospective cohort studies: the English Longitudinal Study of Ageing (ELSA) and the Japan Gerontological Evaluation Study (JAGES). The ELSA targets independent-living older adults aged ≥50, while JAGES participants are community-dwelling individuals aged ≥65 who are ineligible for long-term healthcare insurance benefits.[25] Details of the ELSA and JAGES can be found elsewhere.[26 27] For the present analysis, we used the two waves of data that most closely corresponded with the timing of our study: wave 5 (2010/2011) to wave 6 (2012/2013) for the ELSA, and wave 1 (2010/2011) to wave 2 (2013) for the JAGES. We harmonised the data by including older adults aged ≥65, independent in activities of daily living, and without self-reported dementia. For analysis, respondents who scored above the cut-off point for depression on each measure in the respective cohort at baseline were excluded and we followed up the onset of depression for 2 years for the ELSA and 2.5 years for the JAGES.

### Depression
Based on a previous cross-national study,[28] depressive symptoms were measured both at baseline and follow-up using eight items from the Centre for Epidemiologic Studies Depression Scale (CES-D 8) in the ELSA[29] and the Geriatric Depression Scale (GDS-15) in the JAGES.[30] To identify possible depressive cases, the CES-D 8 cut-off was ≥4 while that for the GDS-15 was ≥5.[31 32] As previously mentioned, respondents with depression at baseline were excluded and we observed the onset of depression during the follow-up.

### Social isolation
Social isolation levels were assessed using a modified version of the Social Isolation Index (SII).[33–35] The index was computed with respondents given a point if they: (1) were unmarried or living alone, (2) had poor interaction with children (did not live with their children or had no one to provide emotional or instrumental social support), (3) had poor interaction with relatives (did not have immediate family members providing emotional or instrumental social support), (4) had poor interaction with friends (less than monthly contact or no friends who could provide emotional or instrumental social support)

and (5) had no social participation (no participation in any social or religious groups). The total possible score ranged from 0 to 5, with higher scores indicating greater social isolation. The participants were categorised into the following five groups based on their scores: 0, 1, 2, 3 and 4–5 points. We used the total score and the scores of the five subcomponents as predictive variables.

## Covariates

The covariates included age, gender, educational attainment, household equivalised income, present illness, self-rated health, smoking and drinking. Age was categorised as 65–69, 70–74, 75–79, 80–84 and ≥85. Based on the ages of respondents who had completed formal education, the age of final educational attainment was categorised as ≤15 years, 16–18 years and ≥19 years. Household equivalised income was classified into quintiles. Present illness was classified as 'yes' or 'no' for cancer, heart disease and stroke. Self-rated health was dichotomised as 'poor' and 'good'. Smoking and drinking were dichotomised as 'never/past' and 'current'.

## Statistical analysis

We analysed the ELSA and JAGES data separately because of differences in research design, especially sampling approaches. A longitudinal weight was applied to account for survey non-response for the ELSA but not the JAGES as its design does not allow it. First, we calculated descriptive statistics. Second, we conducted a multivariable logistic regression analysis to examine the association between SII score and depression onset and obtained ORs and 95% CIs for depression onset. Model 1 was not adjusted for covariates while Model 2 was adjusted for all covariates. Additionally, we analysed the association between SII subcomponents and depression onset, adjusted for all covariates.

To mitigate potential biases resulting from missing information, we used the multiple imputation approach under the missing at random assumption. We generated 20 imputed datasets for the final analysis, which excluded those who met the exclusion criteria and did not respond to the follow-up surveys, using the multiple imputation by chained equations procedure and pooled the results using Rubin's rule.[36]

The significance level was set at P<0.05. We used R (V.3.5.2 for Windows) for all statistical analyses.

## Patient and public involvement

No patients were involved in the development of the research question, study design or data interpretation.

## RESULTS

A total of 3331 ELSA respondents and 33 127 JAGES respondents were included in the final analysis. Their baseline characteristics are presented in table 1. The mean age (SD) was 73.6 (6.9) years for the ELSA and 72.4 (5.4) years for the JAGES. Regarding SII scores, the ELSA had

the largest number of respondents with 0 and 1 points, while the JAGES had the largest number with 2 and 3 points. In the ELSA, respondents who were older, male, less educated, had a lower income, had heart disease, had poor self-rated health, smoked, consumed little alcohol and had higher baseline depressive symptom scores and higher SII scores. A similar trend was observed in the JAGES, but here, those who consumed more alcohol had higher SII scores.

Table 2 presents the description of social isolation and depression onset. At follow-up, 201 (6.0%) ELSA respondents and 4456 (13.5%) JAGES respondents exhibited depression onset. In both studies, higher SII scores were associated with an increased risk of depression onset. Regarding SII subcomponents, ELSA respondents who were unmarried or living alone were more likely to have depression, while this was the case with JAGES respondents with no social participation.

Table 3 depicts the association between SII scores and depression onset. Multivariable analysis showed that higher SII scores were associated with a higher risk of depression onset in both studies after adjusting for all covariates. In the ELSA, the OR of depression onset was significantly higher from a score ≥1 point (OR [95% CI] compared with zero points, one: 1.68 [1.02 to 2.75], two: 1.77 [1.03 to 3.05], three: 2.64 [1.37 to 5.12], ≥four: 4.01 [1.43 to 11.22], P for trend=0.015). In the JAGES, as SII scores increased, the OR of depression onset gradually increased, reaching significance at ≥three points (OR [95% CI] compared with zero points, one: 1.10 [0.89 to 1.35], two: 1.15 [0.94 to 1.40], three: 1.28 [1.04 to 1.56], ≥four: 1.48 [1.18 to 1.85], P for trend <0.001). These results showed almost the same tendency as the complete case analysis without multiple imputation (online supplemental table 1).

Table 4 presents the associations of SII subcomponents with depression onset. In the ELSA, subcomponents were not significant, although poor interaction with children was marginally significant (OR [95% CI]) with 'none' as the reference; unmarried or living alone: 1.13 [0.80 to 1.60], poor interaction with children: 1.55 [1.00 to 2.41], poor interaction with relatives: 1.24 [0.79 to 1.94], poor interaction with friends: 1.15 [0.77 to 1.71], no social participation: 1.22 [0.80 to 1.87]). In the JAGES, poor interaction with children and no social participation were significantly associated with depression onset after adjusting for all covariates (OR [95% CI], with 'none' as the reference; unmarried or living alone: 1.11 [1.00 to 1.24], poor interaction with children: 1.09 [1.01 to 1.19], poor interaction with relatives: 1.04 [0.96 to 1.12], poor interaction with friends: 1.03 [0.95 to 1.11], no social participation: 1.28 [1.17 to 1.40]). These results were similar to those obtained from the complete case analysis (online supplemental table 2).

## DISCUSSION

To the best of our knowledge, this is the first cross-national longitudinal study of the association of social isolation with depression among older English and Japanese adults. Social

**Table 1**  Respondents' baseline characteristics

| | ELSA* | | | | | JAGES | | | | |
|---|---|---|---|---|---|---|---|---|---|---|
| | Social Isolation Index score† | | | | | Social Isolation Index score† | | | | |
| | 0 | 1 | 2 | 3 | ≥4 | 0 | 1 | 2 | 3 | ≥4 |
| | n=905 (27.2%) | n=1049 (31.5%) | n=596 (17.9%) | n=216 (6.5%) | n=49 (1.5%) | n=1402 (4.2%) | n=5981 (18.0%) | n=9723 (29.4%) | n=8735 (26.4%) | n=2176 (6.6%) |
| Age (years), (%) | | | | | | | | | | |
| 65–69 | 41.0 | 36.4 | 29.9 | 27.6 | 28.3 | 38.6 | 40.5 | 38.0 | 37.0 | 35.8 |
| 70–74 | 26.1 | 28.3 | 28.8 | 29.3 | 23.6 | 32.1 | 32.5 | 31.6 | 31.0 | 30.4 |
| 75–79 | 20.9 | 17.5 | 19.3 | 15.8 | 20.4 | 19.2 | 18.3 | 19.5 | 20.3 | 21.9 |
| 80–84 | 9.1 | 12.2 | 12.5 | 14.5 | 15.5 | 8.3 | 6.7 | 8.3 | 9.0 | 8.8 |
| ≥85 | 2.8 | 5.7 | 9.5 | 12.8 | 12.2 | 1.9 | 2.0 | 2.6 | 2.7 | 3.1 |
| Gender, (%) | | | | | | | | | | |
| Men | 50.0 | 46.1 | 45.3 | 51.9 | 65.8 | 27.2 | 35.1 | 47.0 | 64.4 | 66.1 |
| Women | 50.0 | 53.9 | 54.7 | 48.1 | 34.2 | 72.8 | 64.9 | 53.0 | 35.6 | 33.9 |
| Educational attainment (years), (%) | | | | | | | | | | |
| ≤15 | 44.1 | 51.3 | 52.3 | 60.9 | 69.6 | 42.9 | 38.1 | 37.9 | 38.7 | 48.6 |
| 16–18 | 35.0 | 33.8 | 34.6 | 27.5 | 14.1 | 40.9 | 40.8 | 39.3 | 36.8 | 31.1 |
| ≥19 | 17.8 | 12.5 | 11.0 | 10.3 | 14.0 | 15.7 | 20.4 | 21.8 | 23.3 | 18.5 |
| Missing | 3.2 | 2.4 | 2.1 | 1.3 | 2.3 | 0.5 | 0.7 | 0.9 | 1.2 | 1.8 |
| Household equivalised income, (%) | | | | | | | | | | |
| First quintile (lowest) | 11.4 | 18.3 | 23.2 | 31.2 | 14.4 | 7.4 | 10.9 | 13.9 | 15.6 | 21.3 |
| Second quintile | 21.5 | 23.7 | 26.1 | 24.1 | 29.6 | 13.2 | 13.5 | 15.0 | 15.5 | 16.4 |
| Third quintile | 20.9 | 21.8 | 18.9 | 19.5 | 27.1 | 22.5 | 30.6 | 30.3 | 30.6 | 28.0 |
| Fourth quintile | 21.9 | 19.9 | 18.1 | 16.5 | 16.6 | 14.3 | 13.6 | 12.7 | 11.3 | 9.5 |
| Fifth quintile (highest) | 22.7 | 15.0 | 12.7 | 8.3 | 12.4 | 31.5 | 22.5 | 19.2 | 16.8 | 12.9 |
| Missing | 1.5 | 1.3 | 0.9 | 0.4 | 0.0 | 11.1 | 8.9 | 8.9 | 10.2 | 11.9 |
| Cancer, (%) | | | | | | | | | | |
| No | 96.5 | 95.8 | 97.1 | 97.5 | 96.5 | 91.0 | 91.2 | 90.8 | 90.1 | 89.7 |
| Yes | 3.4 | 4.2 | 2.9 | 2.5 | 3.5 | 3.2 | 3.1 | 3.1 | 3.4 | 3.7 |
| Missing | 0.1 | 0.0 | 0.0 | 0.0 | 0.0 | 5.8 | 5.7 | 6.1 | 6.5 | 6.6 |
| Heart disease, (%) | | | | | | | | | | |
| No | 90.7 | 86.9 | 88.4 | 88.3 | 77.3 | 91.0 | 91.2 | 90.8 | 90.1 | 89.7 |
| Yes | 9.3 | 13.0 | 11.6 | 11.7 | 22.7 | 3.2 | 3.1 | 3.1 | 3.4 | 3.7 |
| Missing | 0.0 | 0.1 | 0.0 | 0.0 | 0.0 | 5.8 | 5.7 | 6.1 | 6.5 | 6.6 |
| Stroke, (%) | | | | | | | | | | |
| No | 96.3 | 96.1 | 97 | 94.4 | 96.6 | 93.3 | 93.6 | 93.1 | 92.5 | 92.2 |
| Yes | 3.7 | 3.8 | 3.0 | 5.6 | 3.4 | 0.9 | 0.8 | 0.8 | 1.0 | 1.2 |
| Missing | 0 | 0.1 | 0.0 | 0.0 | 0.0 | 5.8 | 5.7 | 6.1 | 6.5 | 6.6 |
| Self-rated health, (%) | | | | | | | | | | |
| Good | 86.4 | 79.9 | 78.8 | 75.4 | 71.3 | 92.0 | 91.7 | 90.6 | 90.2 | 87.2 |
| Poor | 13.6 | 20.1 | 21.2 | 24.6 | 28.7 | 7.2 | 7.7 | 8.6 | 9.0 | 12.2 |
| Missing | 0.0 | 0.0 | 0.0 | 0.0 | 0.0 | 0.8 | 0.5 | 0.8 | 0.8 | 0.6 |
| Smoking, (%) | | | | | | | | | | |
| Never/past | 95.5 | 93.1 | 89.8 | 82.9 | 89.4 | 86.4 | 86.1 | 82.9 | 81.5 | 78.4 |
| Current | 4.5 | 6.9 | 10.2 | 17.1 | 10.6 | 6.4 | 7.4 | 9.5 | 11.3 | 14.2 |

Continued

**Table 1**  Continued

| | ELSA* | | | | | JAGES | | | | |
|---|---|---|---|---|---|---|---|---|---|---|
| | Social Isolation Index score† | | | | | Social Isolation Index score† | | | | |
| | 0 | 1 | 2 | 3 | ≥4 | 0 | 1 | 2 | 3 | ≥4 |
| | n=905 (27.2%) | n=1049 (31.5%) | n=596 (17.9%) | n=216 (6.5%) | n=49 (1.5%) | n=1402 (4.2%) | n=5981 (18.0%) | n=9723 (29.4%) | n=8735 (26.4%) | n=2176 (6.6%) |
| Missing | 0.0 | 0.0 | 0.0 | 0.0 | 0.0 | 7.1 | 6.5 | 7.5 | 7.2 | 7.3 |
| Drinking, (%) | | | | | | | | | | |
| Never/past | 7.5 | 11.8 | 14.9 | 25.8 | 17.3 | 65.0 | 60.0 | 56.5 | 50.4 | 55.0 |
| Current | 91.3 | 87.0 | 82.9 | 72.6 | 73.6 | 30.2 | 35.7 | 38.6 | 44.7 | 40.4 |
| Missing | 1.3 | 1.2 | 2.2 | 1.6 | 9.2 | 4.8 | 4.3 | 4.9 | 4.9 | 4.6 |
| CES-D 8 score at baseline, (%) | | | | | | | | | | |
| 0 | 57.2 | 52.0 | 45.5 | 42.2 | 54.0 | | | | | |
| 1 | 27.3 | 26.2 | 28.6 | 31.8 | 25.5 | | | | | |
| 2 | 9.8 | 13.8 | 17.1 | 11.9 | 15.1 | | | | | |
| 3 | 5.7 | 8.0 | 8.8 | 14.2 | 5.4 | | | | | |
| GDS score at baseline, (%) | | | | | | | | | | |
| 0 | | | | | | 20.5 | 30.4 | 29.0 | 26.4 | 24.3 |
| 1 | | | | | | 25.5 | 28.4 | 28.6 | 26.9 | 27.0 |
| 2 | | | | | | 22.5 | 20.0 | 19.8 | 21.5 | 21.1 |
| 3 | | | | | | 17.5 | 13.1 | 13.5 | 14.7 | 15.9 |
| 4 | | | | | | 14.1 | 8.1 | 9.1 | 10.5 | 11.7 |

*ELSA data after sampling weight.
†Missing data: ELSA, n=516; JAGES, n=5110.
CES-D 8, eight items from the Centre for Epidemiologic Studies Depression Scale; ELSA, English Longitudinal Study of Ageing; GDS, Geriatric Depression Scale; JAGES, Japan Gerontological Evaluation Study.

isolation was significantly associated with depression onset in both countries. Our results support previous longitudinal findings on social relationships and mental health among older adults in England[37] and Japan.[38] Using data frames that were similar with regard to assessment and covariates, we demonstrated that social isolation is a common risk factor for depression in England and Japan, despite country-specific cultural differences regarding social relationships. Thus, our results suggest that to safeguard the mental health of older adults, addressing social isolation is a global need.

The association between social isolation and depression was somewhat stronger in England than in Japan. These results are similar to a previous report concerning mortality among older adults in England and Japan.[11] Although we cannot make direct comparisons due to variations in cohort follow-up periods and depression measurement, there are several possible reasons for this pattern of findings. The impact of social factors could differ depending on the group and society to which one belongs. This is best understood in the context of the concept of relative deprivation.[39] In other words, higher levels of relative social isolation may induce greater psychological stress. A previous study showed that rich community ties and cohesion were protective factors for health but could have a negative effect on those who were not socially involved.[40] Being isolated in a connected society

such as the UK may represent a more severe condition, with a stronger negative impact on mental health.

Our results revealed that poor interaction with children was significant with regard to depression onset in Japan. In England, while the association was marginal, of the components of social isolation, poor interaction with children had the greatest effect. The lack of interaction with children could have an adverse effect on the mental health of older adults in both countries. Previous studies in England[41] and Japan[22] have reported that social support from children can contribute to alleviating depression, and our results point in the same direction. Older adults without children can be considered a vulnerable group, because adult children, in particular, are often the main source of positive social support for older parents.[42] Older parents have certain expectations with regard to receiving support from their children, and situations wherein these expectations are not met may lead to depressive mood.[43] However, a previous study reported no association between the presence of children and depression among older adults in the USA.[22] Owing to strong spousal relationships in the USA, the effect of the presence of children might be relatively small. Thus, our study confirmed the adverse effects of poor interaction with children common to England and Japan, but international generalisability can only be established based on further research considering

**Table 2**  Description of social isolation status and depression onset

| | ELSA* | | JAGES | |
| --- | --- | --- | --- | --- |
| | CES-D 8 score at follow-up | | GDS score at follow-up | |
| | <4 | ≥4 | <5 | ≥5 |
| | n=3130 (94.0%) | n=201 (6.0%) | n=28 671 (86.5%) | n=4456 (13.5%) |
| Social Isolation Index score, (%) | | | | |
| 0 | 27.5 | 13.5 | 4.4 | 3.0 |
| 1 | 31.2 | 29.3 | 18.6 | 14.3 |
| 2 | 17.9 | 18.9 | 29.8 | 26.3 |
| 3 | 6.5 | 10.1 | 26.2 | 27.5 |
| ≥4 | 1.5 | 2.9 | 6.1 | 9.3 |
| Missing | 15.4 | 25.3 | 14.8 | 19.7 |
| Social Isolation Index sub-components, (%) | | | | |
| Unmarried or living alone | 71.9 | 58.5 | 88.3 | 85.2 |
| No | | | | |
| Yes | 28.1 | 41.5 | 10.2 | 12.7 |
| Missing | 0.0 | 0.0 | 1.4 | 2.1 |
| Poor interaction with children | 81.8 | 73.4 | 26.6 | 25.0 |
| No | | | | |
| Yes | 12.5 | 14.7 | 71.7 | 72.9 |
| Missing | 5.7 | 11.8 | 1.7 | 2.1 |
| Poor interaction with relatives | 76.3 | 68.1 | 41.1 | 38.7 |
| No | | | | |
| Yes | 17.3 | 18.6 | 54.4 | 56.0 |
| Missing | 6.4 | 13.4 | 4.5 | 5.4 |
| Poor interaction with friends | 72.2 | 66.0 | 37.1 | 31.2 |
| No | | | | |
| Yes | 18.4 | 21.6 | 58.2 | 63.8 |
| Missing | 9.4 | 12.4 | 4.6 | 5.0 |
| No social participation | 61.7 | 46.3 | 75.0 | 63.9 |
| No | | | | |
| Yes | 28.4 | 32.5 | 13.0 | 20.0 |
| Missing | 10.0 | 21.2 | 11.7 | 16.1 |

*ELSA data after sampling weight.
CES-D 8, eight items from the Centre for Epidemiologic Studies Depression Scale; ELSA, English Longitudinal Study of Ageing; GDS, Geriatric Depression Scale; JAGES, Japan Gerontological Evaluation Study.

the cultural background of family relationships in individual countries.

Although traditionally Japan is a country in which adult children are expected to demonstrate reciprocity with their parents based on the strong family and kinship-based cultural background,[44] in this study, the effect of interaction with children on depression was relatively modest. In recent years, with trends such as adult children commonly living apart from their parents after marriage[45] and the development of public long-term care services for the ageing population,[46] Japan's family system has become less traditional. Therefore, interaction with children may not be as essential to the health of older adults as before. However, despite these cultural transitions, we believe that interaction with children has some value with regard to preventing depression in old age in Japan.

Social participation was a strong protective factor for depression onset in Japan, whereas there was no association in England, although the OR was somewhat greater. Several previous studies have reported that social participation helps prevent depression onset.[37 47–49] Our results pertaining to Japan support these reports. However, the protective effects of social participation on mental health vary depending on the type of organisation with which an individual is involved,[48]

**Table 3** Association between social isolation and depression onset: multivariable logistic regression analysis

| | ELSA | | JAGES | |
|---|---|---|---|---|
| | Crude OR (95% CI) | Adjusted OR (95% CI) | Crude OR (95% CI) | Adjusted OR (95% CI) |
| Social Isolation Index score | | | | |
| 0 | 1.00 (reference) | 1.00 (reference) | 1.00 (reference) | 1.00 (reference) |
| 1 | 1.92** (1.19 to 3.10) | 1.68* (1.02 to 2.75) | 1.14 (0.94 to 1.39) | 1.10 (0.89 to 1.35) |
| 2 | 2.15** (1.28 to 3.62) | 1.77* (1.03 to 3.05) | 1.32** (1.09 to 1.60) | 1.15 (0.94 to 1.40) |
| 3 | 3.19*** (1.73 to 5.90) | 2.64** (1.37 to 5.12) | 1.57*** (1.30 to 1.90) | 1.28* (1.04 to 1.56) |
| ≥4 | 3.85** (1.46 to 10.18) | 4.01** (1.43 to 11.22) | 2.26*** (1.83 to 2.79) | 1.48*** (1.18 to 1.85) |
| | P for trend <0.001 | P for trend=0.015 | P for trend <0.001 | P for trend <0.001 |

Adjusted for age, gender, educational attainment, household equivalised income, present illness (cancer, heart disease, and stroke), self-rated health, smoking, drinking, and depression score at baseline (eight items from the Centre for Epidemiologic Studies Depression Scale for the ELSA and Geriatric Depression Scale for the JAGES).
*P<0.05; **P<0.01; ***P<0.001.
ELSA, English Longitudinal Study of Ageing; JAGES, Japan Gerontological Evaluation Study.

the individual's attitude towards participation,[48] and the duration[37] and frequency[49] of participation. Regarding the present study, in the English context, the role of social participation in depression prevention might have been unidentifiable due to differences in the effects of these participation contexts. We took into account only social participation, without delving into specific types. Thus, the context of effective social participation, such as type, duration, and role in the organisation in both countries, requires further investigation. Despite these challenges, our findings suggest that in Japan, social isolation prevention measures based on the promotion of social participation could be beneficial for safeguarding the mental health of older adults.

This study has several strengths. First, it is the first cross-national population-level investigation of the association of social isolation with depression onset using a unified data frame. Second, by using two longitudinal datasets, we were able to examine the prospective association between social isolation and depression. Third, the use of large-scale data allowed us to detect the effects of relatively rare situations of severe social isolation.

However, certain limitations cannot be ignored. First, the measurement of depression in the two cohorts was not the same. Therefore, we could not directly compare depression onset in the two countries. However, these measurements were also used in a previous cross-national comparison study in England and Japan,[28] and we were able to examine the association between social isolation and depression onset in both countries using the same data frame. Second, we used social support for the assessment of social contact for some items in order to be able to use the same SII. Therefore, cultural differences in expectations regarding the receipt of social support in both countries might have caused information biases. For instance, expectations regarding social support from relatives could originally have been higher in Japan,[44] leading to overestimation of social isolation levels. Third, regarding the items of the SII, the questions and their response options in the ELSA and JAGES were not exactly the same, nor were they strictly authorised through procedures such as reverse translation and confirming reliability and validity. However, we believe it is certainly meaningful to evaluate social isolation using the same framework. Finally,

**Table 4** Association between subcomponents of social isolation and depression onset: multivariable logistic regression analysis

| | ELSA | JAGES |
|---|---|---|
| | Adjusted OR (95% CI) | Adjusted OR (95% CI) |
| Social Isolation Index subcomponents (reference: none) | | |
| Unmarried or living alone | 1.13 (0.80 to 1.60) | 1.11† (1.00 to 1.24) |
| Poor interaction with children | 1.55† (1.00 to 2.41) | 1.09* (1.01 to 1.19) |
| Poor interaction with relatives | 1.24 (0.79 to 1.94) | 1.04 (0.96 to 1.12) |
| Poor interaction with friends | 1.15 (0.77 to 1.71) | 1.03 (0.95 to 1.11) |
| No social participation | 1.22 (0.80 to 1.87) | 1.28*** (1.17 to 1.40) |

Adjusted for age, gender, educational attainment, equivalent income, present illness (cancer, heart disease and stroke), self-rated health, smoking, drinking and depression score at baseline (eight items from the Centre for Epidemiologic Studies Depression Scale for the ELSA and Geriatric Depression Scale for the JAGES).
*P<0.05; ***P<0.001;†P<0.1.
ELSA, English Longitudinal Study of Ageing; JAGES, Japan Gerontological Evaluation Study.

there were differences in study design in the data from the two cohorts, such as sampling method and follow-up period. We, therefore, made efforts to harmonise the data: those aged ≤64, with dementia, and dependent in activities of daily living were excluded from the analysis. Moreover, the ELSA presents nationally representative population data, while the JAGES does not. However, the JAGES sample is representative of areas from a nationwide ageing study in which approximately one-fifth of all prefectures (9 out of 47) were enrolled. Even so, unlike the ELSA, analysis in the JAGES does not use sampling weights, which may lead to selection bias.

## CONCLUSION

We examined the association between social isolation and depression onset among older adults in England and Japan, who experience different cultural contexts regarding social relationships, and found a significant association in both countries; we also observed that in England, poor interaction with children was marginally associated, and in Japan, poor interaction and lack of social participation were significantly associated with depression. Tackling social isolation must be prioritised to safeguard the mental health of older adults worldwide. Particularly in Japan, the promotion of interaction with children and social participation could be key factors in addressing social isolation.

### Author affiliations
[1]Department of Social Science, Center for Gerontology and Social Science, National Center for Geriatrics and Gerontology, Obu, Japan
[2]Department of Public Health, Nagoya City University Graduate School of Medical Sciences and Medical School, Nagoya, Japan
[3]Faculty of Social Welfare, Nihon Fukushi University, Chita-gun, Japan
[4]Department of Oral Health Promotion, Tokyo Medical and Dental University Graduate School of Medical and Dental Sciences, Bunkyo-ku, Japan
[5]Division for Regional Community Development, Liaison Center for Innovative Dentistry, Tohoku University, Sendai, Japan
[6]Department of Epidemiology and Public Health, University College London, London, UK
[7]Faculty of Health and Sport Sciences, University of Tsukuba, Tokyo, Japan
[8]Department of Social Preventive Medical Sciences, Center for Preventive Medical Sciences, Chiba University, Chiba, Japan
[9]Cancer Control Center, Osaka International Cancer Institute, Osaka, Japan
[10]Department of Health Policy Science, Yamagata University Faculty of Medicine Graduate School of Medical Science, Yamagata, Japan
[11]Department of International and Community Oral Health, Tohoku University Graduate School of Dentistry School of Dentistry, Sendai, Japan
[12]Department of Gerontological Evaluation, Center for Gerontology and Social Science, National Center for Geriatrics and Gerontology, Obu, Japan

**Acknowledgements** We wish to express our deepest gratitude to the members in the Japan Gerontological Evaluation Study. We would also like to thank everyone who participated in the surveys.

**Contributors** All authors contributed to the conception and design of this study. Data collection was primarily conducted by MS, JA, NC, KO and KK. Analyses were performed by TN, MS, JA, TT, SK and TI. TN prepared the initial manuscript and MS, JA, NC, TT, SK, TI, KO, and KK significantly contributed to revising it. All authors read and approved the final manuscript.

**Funding** This work was supported by a grant from the Grants-in-Aid for Scientific Research (18KK0057, 19K24277) from the Japan Society for the Promotion of Science. The English Longitudinal Study of Ageing was developed by a team of researchers based at the University College London, NatCen Social Research, and the Institute for Fiscal Studies. The data were collected by NatCen Social Research. The funding is currently provided by the National Institute of Aging (R01AG017644) and a consortium of UK government departments coordinated by the National Institute for Health Research. The Japan Gerontological Evaluation Study is supported by the Ministry of Education, Culture, Sports, Science and Technology-Japan-Supported Program for the Strategic Research Foundation at Private Universities (2009-2013), Japan Society for the Promotion of Science KAKENHI Grant Numbers (JP18390200, JP22330172, JP22390400, JP23243070, JP23590786, JP23790710, JP24390469, JP24530698, JP24683018, JP25253052, JP25870573, JP25870881, JP26285138, JP26882010, JP15H01972), Health Labour Sciences Research Grants (H22-Choju-Shitei-008, H24-Junkanki -Ippan-007, H24-Chikyukibo-Ippan-009, H24-Choju-Wakate-009, H25-Kenki-Wakate-015, H25-Choju-Ippan-003, H26-Irryo-Shitei-003 [Fukkou], H26-Choju-Ippan-006, H27-Ninchisyou-Ippan-001, H28-chojulppan-002, H28-Ninchisho-Ippan-002, H30-Kenki-Ippan-006, H30-Junkankitou-Ippan-004), Japan Agency for Medical Research and Development (JP17dk0110017, JP18dk0110027, JP18ls0110002, JP18le0110009, JP19dk0110034), Research Funding for Longevity Sciences from the National Center for Geriatrics and Gerontology (24-17, 24-23, 29-42, 30-22, 20-40), Open Innovation Platform with Enterprises, Research Institute and Academia (OPERA, JPMJOP1831) from the Japan Science and Technology Agency, and a Research Grant from the Health Science Center Foundation (2019-2020). The funding sources had no role in the study design, data collection and analysis, decision to publish or preparation of the manuscript.

**Competing interests** None declared.

**Patient consent for publication** Not required.

**Ethics approval** The ELSA investigators received ethical approval for all waves of the study from the National Health Service Research Ethics Committees under the National Research and Ethics Services (MREC/01/2/91). The JAGES protocols were approved by the Ethics Committee on the Research of Human Subjects at Nihon Fukushi University (10-05).

**Provenance and peer review** Not commissioned; externally peer reviewed.

**Data availability statement** Data may be obtained from a third party and are not publicly available. For the JAGES, all enquiries are to be addressed to the data management committee via email: dataadmin.ml@jages.net. All JAGES datasets have ethical or legal restrictions for public deposition because of the inclusion of sensitive information about the human participants. Regarding the ELSA, data are available in an open-access repository at https://www.data-archive.ac.uk/.

### ORCID iDs
Taiji Noguchi http://orcid.org/0000-0001-9165-5501
Masashige Saito http://orcid.org/0000-0002-3997-3884
Jun Aida http://orcid.org/0000-0002-8405-9872
Noriko Cable http://orcid.org/0000-0001-5478-1760
Taishi Tsuji http://orcid.org/0000-0002-8408-6619
Shihoko Koyama http://orcid.org/0000-0002-7786-4910
Takaaki Ikeda http://orcid.org/0000-0003-4325-4492
Ken Osaka http://orcid.org/0000-0001-6885-9514
Katsunori Kondo http://orcid.org/0000-0003-0076-816X

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
