## [Reviewer comments · BMJ Open]

ARTICLE DETAILS

TITLE (PROVISIONAL)	Association between social isolation and depression onset among older adults: A cross-national longitudinal study in England and Japan
AUTHORS	Noguchi, Taiji; Saito, Masashige; Aida, Jun; Cable, Noriko; Tsuji, Taishi; Koyama, Shihoko; Ikeda, Takaaki; Osaka, Ken; Kondo, Katsunori

VERSION 1 – REVIEW

REVIEWER	Ziggi Santini University of Southern Denmark, Denmark
REVIEW RETURNED	27-Nov-2020

GENERAL COMMENTS	Introduction: Page 6, line 89-91: The cited study on mortality shows that these differences in friend and family-based relationships impacted survival only for men, not women. This needs to be mentioned. Page 7, line 92-94: The authors state that the cited study found that the presence of children was associated with depressive symptoms in Japan. However, what the study actually found was that the presence of children was associated with fewer depressive symptoms, which is a very important (opposite) difference. Also, the cited study also found that the effect of spousal presence was stronger in the US than in Japan, which would seem to go against the theory of Japanese culture being more family-based (if spouses are considered family, which I assume). Additionally, the cited study found that the effect of the presence of children in Japan was much stronger among those not married, which would suggest that children are particularly important in Japan among those that are otherwise alone (without a spouse). How does all this fit into the theory the authors are arguing for? https://spssi.onlinelibrary.wiley.com/doi/abs/10.1111/1540-4560.00290 Overall, if the authors want to maintain that UK is more friend-based and Japan is more family-based, they need to provide much more compelling evidence than they have. Both friends and family probably matter in both contexts, so it's hard to argue that one is necessarily more oriented towards one thing than another. Page 7, line 102: Please define "super-aged society". And why is Japan seeing an increase in abstaining from marriage? Abstinence from marriage, when it happens, is usually a religious matter. Page 7, line 108: Why is the impact of social isolation on depression symptoms expected to vary across countries? The study so far talks about differences in terms of prevalence of social isolation between Japan and the UK, but why would the association between social isolation and depression be any different in the UK than in Japan? Is social isolation itself expected to be more detrimental in Japan than in the UK? If so, please state why. Page 8, line 111: Longitudinal observational studies are stronger than cross-sectional studies, but they still cannot establish causality. You can still have a third variable-problem. Methods:
--

	Page 9, line 133-138: Why were respondents with any number of depressive symptoms excluded? Why not use the cut-points on the CESD and the GDS-15 to exclude only those screening positive for clinically significant depression at baseline? Depression symptoms are not a problem unless a person experiences so much of it that it impairs daily functioning. What the authors are doing at the moment is restricting the data to those with perfect mental health (i.e. no symptoms at all), which is not very realistic. Most people experience ups and downs in life without having depression. This is a normal part of life. The standard approach would be to exclude those with depression case at baseline, thereby isolating the population at risk for depression. Similar to below study: https://www.sciencedirect.com/science/article/pii/S0165032715314142?via%3Dihub Page 9, line 140-145: The items used for the SSI, where they based on exactly the same items and response options in the UK and Japan surveys? Page 15, line 206-212: The authors are not assessing descriptives for “depressive symptoms” onset, but rather “depression onset”. They did not assess descriptives or risk for any number of symptoms, but rather enough symptoms to qualify for clinically significant depression. This should be reflected throughout the paper where relevant. I.e. change from “depressive symptoms” to “depression”. Discussion: Page 22, line 277: The authors state that poor friend support was not associated with depression, however, the OR is 1.15, when suggest an association, albeit small. Rather, there is a small association, but it did not reach statistical significance at the 95% level. Also, results for ELSA indicate that support from family and relatives is more important than friend support (although non-significant but still larger ORs), as well as the cited study #32. Does this not conflict with the theory of UK culture being more friendship-based? Page 23, line 295: Again, it is not possible to determine causality with observational data.
--	--

REVIEWER	Namkee G Choi University of Texas at Austin
REVIEW RETURNED	29-Nov-2020

GENERAL COMMENTS	This is an interesting study that examined the impact of social isolation on the onset of depression among older adults in England and Japan. Given the interest in social isolation and loneliness in late life in recent years (and especially with Covid-19), the study findings will add to the knowledge base. There are several areas that need clarifications:  1. From the outset, the authors emphasized cultural difference between two countries—importance of friend-based social interactions in England and family-based social interactions in Japan. However, the results do not support these cultural differences in relation to depressive symptoms. In both countries, poor interactions with children were found to be a more important risk factor than poor interactions with friends. However, the authors did not really discuss the discrepancy between their initial conceptualization and the findings. Rather than data limitations as a possible cause, they should delve more into the universal importance of poor social interactions with children (but not marital status/living arrangement and lack of social support from other relatives). The author should look up previous research on depression if they identified poor interaction with children as a risk factor for late-life depression regardless of nationalities. 2. The authors also need to have more in-depth discussion of the importance of no social participation in Japan but not in England. Previous research shows that social engagement and participation in late life is an important depression protective factor. If so, why lack of social participation is not a risk factor for older adults in England? In sum, I think the Discussion section is the weakest link
--

	in this paper. The authors need to have more thoughtful discussion. 3. Minor comments: a. Page 7, line 93: "... presence of children was associated with depression...": Please clarify this. Does this mean that more children increased depressive symptoms? b. Page 9, lines 143-146: Please clarify "social support": what kind of social support? Emotional? Instrumental? Financial? c. Page 11, line 165: How come the Japanese survey data were not weighted? Did it not use a sampling design to reach a nationally representative sample? If so, shouldn't you state that as a limitation? d. Table 1: please specify % after the number of cases in each SII score group.
--	--

VERSION 1 – AUTHOR RESPONSE

Dear Reviewer 1,
 Dr. Ziggi Ivan Santini, University of Southern Denmark

General response:

We appreciate your thorough evaluation of our manuscript and are grateful for the feedback. We have tried our best to incorporate your suggestions in the revised manuscript, where all changes are highlighted in yellow.

Comment #1:

Page 6, line 89-91: The cited study on mortality shows that these differences in friend and family-based relationships impacted survival only for men, not women. This needs to be mentioned.

Response #1:

Thank you for your advice. Regarding the cited study, we have made it clear that the results pertain solely to men as follows.

(pp 8, lines 80 to 82) A previous cross-national study showed that in English men, friendship-based social relationships had a significant impact on longevity, whereas in Japanese men, this impact was associated with family-based social relationships.[21]

Comment #2:

Page 7, line 92-94: The authors state that the cited study found that the presence of children was associated with depressive symptoms in Japan. However, what the study actually found was that the presence of children was associated with fewer depressive symptoms, which is a very important (opposite) difference. Also, the cited study also found that the effect of spousal presence was stronger in the US than in Japan, which would seem to go against the theory of Japanese culture being more family-based (if spouses are considered family, which I assume). Additionally, the cited study found that the effect of the presence of children in Japan was much stronger among those not married, which would suggest that children are particularly important in Japan among those that are otherwise alone (without a spouse). How does all this fit into the theory the authors are arguing for?
<https://spssi.onlinelibrary.wiley.com/doi/abs/10.1111/1540-4560.00290>

Overall, if the authors want to maintain that UK is more friend-based and Japan is more family-based, they need to provide much more compelling evidence than they have. Both friends and family

probably matter in both contexts, so it's hard to argue than one is necessarily more oriented towards one thing than another.

Response #2:

We appreciate your thoughtful suggestions very much. As you pointed out, our hypothesis, which was that England (a Western country) places importance on friendship-based relationships while Japan (an Eastern country) places importance on family-based relationships, could be considered weak because of insufficient evidence. Therefore, we have modified the hypothesis as follows: differences in the association between social isolation and depression are related to the degree of social connectivity in each country. Essentially, we hypothesised that the effect of social isolation on depression differs because of variations in the social structures in England and Japan.

The effects of social isolation components (family, friends, and others) on depression may also vary from country to country. However, as the results of previous studies are inconsistent, we considered the problems inherent in hypothesising that family is important in Japan and friends are important in England. Therefore, we have mentioned that identifying the specific components of social isolation associated with depression in older adults in both countries is controversial. We have accordingly revised the Introduction as follows.

(pp 7 to 8, lines 73 to 91) Furthermore, the health effects of social isolation may differ depending on the cultural context of social relationships. In East Asian countries, including Japan, there is a familial norm based on the traditional culture of filial piety,[17] which is often contrasted with individualism in Western countries.[18, 19] Based on this cultural background, Japanese social support networks may be kinship centred, which may be narrower than the types of social networks prevalent in other countries.[20] However, there is a lack of consensus on the health effects of social relationships based on these cultural differences. A previous cross-national study showed that in English men, friendship-based social relationships had a significant impact on longevity, whereas in Japanese men, this impact was associated with family-based social relationships.[21] On the contrary, a study of older adults in the United States (US) and Japan found that while relationships with children were associated with a low level of depression only in Japan, the presence of spouses was important in both countries, but more so in the US.[22] Another comparative study among adults suggested that social contact with friends benefitted women's mental health in the UK but not in Japan.[23] Thus, the family-oriented nature of East Asian societies does not automatically imply the health importance of family-based relationships, and the roles of individual components of social isolation (family, friends, and others) in the mental health of older adults in each country remain controversial.

Comment #3:

Page 7, line 102: Please define "super-aged society". And why is Japan seeing an increase in abstaining from marriage? Abstinence from marriage, when it happens, is usually a religious matter.

Response #3:

Thank you for your feedback. We have now clearly stated that a 'super-aged society' is one where more than 21% of the population is aged 65 or above.

The reason for the increase in the number of people abstaining from marriage in Japan is not related to religion; rather, this trend is believed to be the result of rapid improvement in women's education and the accompanying dramatic increase in the number of working women, changes in the structure and function of the marital market (the collapse of the arranged marriage system), an increase in premarital sexual relations, and changes in the values associated with married life (Retherford RD, et al., Population and Development Review, 2004).

(pp 7, lines 63 to 67) In contrast, Japan, now a super-aged society (more than 21% of the population aged 65 or above),^[13] is experiencing a rapid increase in the trend of abstaining from marriage and weakening community and neighbourhood relations,^[14] leading to a rise in the number of socially isolated people.^[15]

Comment #4:

Page 7, line 108: Why is the impact of social isolation on depression symptoms expected to vary across countries? The study so far talks about differences in terms of prevalence of social isolation between Japan and the UK, but why would the association between social isolation and depression be any different in the UK than in Japan? Is social isolation itself expected to be more detrimental in Japan than in the UK? If so, please state why.

Response #4:

Thank you for your important advice. We hypothesised that the differences in social environments related to social networks and relationships across countries affect the impact of social isolation on depression onset. The UK is known for taking advanced measures against social isolation and loneliness, such as the establishment of the position of 'Minister of Loneliness.' On the contrary, as OECD reports have shown, Japan has one of the highest levels of social isolation, with further increases expected. In fact, a study examining the association between social isolation and mortality showed a stronger correlation in the UK than Japan (Saito M, et al., *Geriatr Gerontol Int*, 2020). Therefore, the impact of social isolation on depression was expected to vary across countries owing to differences in social structures and the contexts of social isolation. In consideration of your comments, we have modified the Introduction as follows.

(pp 6 to 7, lines 55 to 72) The impacts of social isolation on health may vary by country; this could be the result of differences in the social environments related to social networks within and outside the family. A recent study of older adults in England and Japan showed that social isolation is a common risk factor for mortality in both countries, with a greater impact in England; the results are discussed in terms of possible differences between societies that are highly connected and those that are not.^[11] In the United Kingdom (UK), in recognition of the impact of social isolation on health and economic loss, the position of 'Minister of Loneliness' was established in 2018, and the country is taking a progressive approach to the elimination of social isolation.^[12] In contrast, Japan, now a super-aged society (more than 21% of the population aged 65 or above),^[13] is experiencing a rapid increase in the trend of abstaining from marriage and weakening community and neighbourhood relations,^[14] leading to a rise in the number of socially isolated people.^[15] In Japan, the proportion of people who rarely or never spend time with those close to them has been reported to be the highest among Organisation for Economic Co-operation and Development countries. In particular, this figure is much higher than in the UK, which has made advances in tackling social isolation (Japan = 15.3%, UK = 5.0%).^[16] Owing to differences in social structures and the contexts surrounding social isolation, the impact of social isolation on depression is expected to vary across countries.

Comment #5:

Page 8, line 111: Longitudinal observational studies are stronger than cross-sectional studies, but they still cannot establish causality. You can still have a third variable-problem.

Response #5:

Thank you for your comments. As you pointed out, causality cannot be determined; therefore, we have revised the description as follows.

(pp 8, lines 92 to 93) As the association between social isolation and depression is often described as bidirectional,[24] longitudinal studies are needed to address temporality.

Comment #6:

Page 9, line 133-138: Why were respondents with any number of depressive symptoms excluded? Why not use the cut-points on the CESD and the GDS-15 to exclude only those screening positive for clinically significant depression at baseline? Depression symptoms are not a problem unless a person experiences so much of it that it impairs daily functioning. What the authors are doing at the moment is restricting the data to those with perfect mental health (i.e. no symptoms at all), which is not very realistic. Most people experience ups and downs in life without having depression. This is a normal part of life. The standard approach would be to exclude those with depression case at baseline, thereby isolating the population at risk for depression. Similar to below study:

<https://www.sciencedirect.com/science/article/pii/S0165032715314142?via%3Dihub>

Response #6:

Thank you for your suggestions. We apologise for the lack of clarity in our original explanation. To be clear, we did not exclude respondents with any number of depressive symptoms at baseline but only those who screened positive based on the cut-off for each measure, that is, those with CES-D scores ≥ 4 in the ELSA and GDS-15 scores ≥ 5 in the JAGES. This approach is similar to the report by Freeman et al. We have now rephrased as follows.

(pp 9 to 10, lines 111 to 114) For analysis, respondents who scored above the cut-off point for depression on each measure in the respective cohort at baseline were excluded, and we followed up the onset of depression for two years for the ELSA and 2.5 years for the JAGES.

(pp 10, lines 121 to 123) As previously mentioned, respondents with depression at baseline were excluded, and we observed the onset of depression during follow-up.

Comment #7:

Page 9, line 140-145: The items used for the SSI, where they based on exactly the same items and response options in the UK and Japan surveys?

Response #7:

Thank you for your comments. Regarding the items of the Social Isolation Index, we used the same assessment items for both countries to ensure comparability. However, the questions and their response options were not exactly the same, and were not strictly authorised through procedures such as reverse translation and confirming reliability and validity. For instance, there is no denying that the perception of receiving social support, as well as the meaning and definition of the term 'friend', may differ depending on the culture of the country. We have added these limitations as follows.

(pp 27, lines 319 to 323) Third, regarding the items of the SII, the questions and their response options in the ELSA and JAGES were not exactly the same, nor were they strictly authorised through procedures such as reverse translation and confirming reliability and validity. However, we believe it is certainly meaningful to evaluate social isolation using the same framework.

Comment #8:

Page 15, line 206-212: The authors are not assessing descriptives for “depressive symptoms” onset, but rather “depression onset”. They did not assess descriptives or risk for any number of symptoms, but rather enough symptoms to qualify for clinically significant depression. This should be reflected throughout the paper where relevant. I.e. change from “depressive symptoms” to “depression”.

Response #8:

Thank you for your suggestions. We agree and have, accordingly, changed ‘depressive symptom onset’ to ‘depression onset’ throughout the manuscript.

Comment #9:

Page 22, line 277: The authors state that poor friend support was not associated with depression, however, the OR is 1.15, when suggest an association, albeit small. Rather, there is a small association, but it did not reach statistical significance at the 95% level. Also, results for ELSA indicate that support from family and relatives is more important than friend support (although non-significant but still larger ORs), as well as the cited study #32. Does this not conflict with the theory of UK culture being more friendship-based?

Response #9:

Thank you for your suggestions. As you pointed out, our results did not underscore the importance of friendship-based relationships with regard to mental health in England. As noted in our response to Comment #2, we have revised the hypothesis that friendship-based relationships are critical to health in Western countries; it now states that the effects of social isolation components on mental health vary from country to country. Therefore, we have corrected the assumption that people in England benefit from friendship-based relationships throughout the manuscript. Under Discussion, we have focused on interaction with children and social participation, which were significant with regard to depression.

Comment #10:

Page 23, line 295: Again, it is not possible to determine causality with observational data.

Response #10:

Thank you for your advice. As it is not possible to determine causality with observational data, we have made the following revision.

(pp 26, lines 305 to 306) Second, by using two longitudinal datasets, we were able to examine the prospective association between social isolation and depression.

Dear Reviewer 2,
Dr. Namkee G. Choi, Baylor Coll Med

General comments:

This is an interesting study that examined the impact of social isolation on the onset of depression among older adults in England and Japan. Given the interest in social isolation and loneliness in late life in recent years (and especially with Covid-19), the study findings will add to the knowledge base. There are several areas that need clarifications:

Response to general comments:

We appreciate your thorough evaluation of our manuscript and the constructive feedback you have provided. We have highlighted all our changes in yellow in the revised manuscript.

Comment #1:

From the outset, the authors emphasized cultural difference between two countries—importance of friend-based social interactions in England and family-based social interactions in Japan. However, the results do not support these cultural differences in relation to depressive symptoms. In both countries, poor interactions with children were found to be a more important risk factor than poor interactions with friends. However, the authors did not really discuss the discrepancy between their initial conceptualization and the findings. Rather than data limitations as a possible cause, they should delve more into the universal importance of poor social interactions with children (but not marital status/living arrangement and lack of social support from other relatives). The author should look up previous research on depression if they identified poor interaction with children as a risk factor for late-life depression regardless of nationalities.

Response #1:

Thank you for your very important advice. It is true that our results did not support our original assumption of the importance of friendship-based relationships in England and family-based relationships in Japan. Therefore, we have revised our hypothesis to reflect the fact that the impact of social isolation on depression in England and Japan varies because of differences in social environments; the UK has made advances against social isolation as represented by the establishment of the position of the ‘Minister of Loneliness,’ whereas Japan is a super-aged society with the largest number of socially isolated people in the world. Regarding the components of social isolation, such as interaction with children, as there was no consistent trend, this served as the second research question.

In the present study, poor interaction with children affected depression onset in both England and Japan. We have discussed these results in depth from the aspect of the meaning of the presence of children in relation to the mental health of older adults. In addition, we have discussed the reasons why the impact was not dramatic in Japan, where family and kinship are considered to be strong, from the viewpoint of recent changes in Japanese family culture.

(pp 24 to 25, lines 261 to 287) Our results showed that poor interaction with children was significant with regard to depression onset in Japan. In England, while the association was marginal, of the components of social isolation, poor interaction with children had the greatest effect. The lack of interaction with children could have an adverse effect on the mental health of older adults in both countries. Previous studies in England[41] and Japan[22] have reported that social support from children can contribute to alleviating depression, and our results point in the same direction. Older adults without children can be considered a vulnerable group, because adult children, in particular, are often the main source of positive social support for older parents.[42] Older parents have certain expectations with regard to receiving support from their children, and situations wherein these expectations are not met may lead to depressive mood.[43] On the contrary, a previous study reported no association between the presence of children and depression among older adults in the US.[22] Owing to strong spousal relationships in the US, the effect of the presence of children might be relatively small. Thus, our study confirmed the adverse effects of poor interaction with children common to England and Japan, but international generalizability can only be established based on further research considering the cultural background of family relationships in individual countries.

Although traditionally Japan is a country where adult children are expected to demonstrate reciprocity with their parents owing to the strong family and kinship-based cultural background,[44] in this study, the effect of interaction with children on depression was relatively modest. In recent years, with trends such as adult children commonly living apart from their parents after marriage[45] and the development of public long-term care services for the ageing population,[46] Japan’s family system has not remained as traditional as before. Therefore, interaction with children may not be as

essential to the health of older adults as before. However, despite these cultural transitions, we believe that interaction with children has some value with regard to preventing depression in old age in Japan.

Comment #2:

The authors also need to have more in-depth discussion of the importance of no social participation in Japan but not in England. Previous research shows that social engagement and participation in late life is an important depression protective factor. If so, why lack of social participation is not a risk factor for older adults in England? In sum, I think the Discussion section is the weakest link in this paper. The authors need to have more thoughtful discussion.

Response #2:

Thank you for your thoughtful advice. As you pointed out, there are several previous studies that show the importance of social participation in preventing depression. We believe that our results pertaining to Japanese older adults support these reports. In England, however, no statistical association was observed, although the odds ratio of no social participation for depression onset was somewhat high. Previous studies have shown that the benefits to mental health could be significant depending on the context of social participation (the type of organisation, attitude towards participation, and duration and frequency of participation). As our understanding of social participation was based on involvement in organisations, we might not have been able to identify the association owing to the differences in the effects of each context. Therefore, we believe that further research is needed to identify the ideal context of social participation among older adults in England. We have revised the Discussion as follows.

(pp 25 to 26, lines 288 to 302) Social participation was a strong protective factor for depression onset in Japan, whereas there was no association in England, although the OR was somewhat greater. Several previous studies have reported that social participation helps prevent depression onset.[37, 47–49] Our results pertaining to Japan support these reports. On the contrary, the protective effects of social participation on mental health have been shown to vary depending on the type of organisation an individual is involved with,[48] the individual's attitude towards participation,[48] and the duration[37] and frequency[49] of participation. Regarding the present study, in the English context, the role of social participation in depression prevention might have been unidentifiable because of differences in the effects of these participation contexts. We only took into account social participation, without delving into specific types. Thus, the context of effective social participation, such as type, duration, and role in the organisation in both countries, requires further investigation. Despite these challenges, our findings suggest that in Japan, social isolation prevention measures based on the promotion of social participation could be beneficial for safeguarding the mental health of older adults.

Comment #3-a:

Page 7, line 93: "... presence of children was associated with depression...": Please clarify this. Does this mean that more children increased depressive symptoms?

Response #3-a:

Thank you for your comments. The cited study showed 'an inverse association between the presence of children and depression'; we apologise for the vague description. We have revised the description of this cited study as follows.

(pp 8, lines 82 to 85) On the contrary, a study of older adults in the United States (US) and Japan found that while relationships with children were associated with a low level of depression only in Japan, the presence of spouses was important in both countries, but more so in the US.[22]

Comment #3-b:

Page 9, lines 143-146: Please clarify “social support”: what kind of social support? Emotional? Instrumental? Financial?

Response #3-b:

Thank you for your comments. We used emotional and instrumental support. This has been clarified as follows.

(pp 10 to 11, lines 127 to 134) The index was computed with respondents given a point if they: (1) were unmarried or living alone, (2) had poor interaction with children (did not live with their children or had no one to provide emotional or instrumental social support), (3) had poor interaction with relatives (did not have immediate family members providing emotional or instrumental social support), (4) had poor interaction with friends (less than monthly contact or no friends who could provide emotional or instrumental social support), and (5) had no social participation (no participation in any social or religious groups).

Comment #3-c:

Page 11, line 165: How come the Japanese survey data were not weighted? Did it not use a sampling design to reach a nationally representative sample? If so, shouldn't you state that as a limitation?

Response #3-c:

Thank you for your advice. As sampling weights were not available for the JAGES, we have included this as a limitation of the study as follows.

(pp 28, lines 329 to 330) Even so, unlike the ELSA, analysis in the JAGES does not use sampling weights, which may lead to selection bias.

Comment #3-d:

Table 1: please specify % after the number of cases in each SII score group.

Response #3-d:

Thank you for your comments. We have added '%' after the number of cases in each SII score group (pp 15, Table 1 and pp 18, Table 2).

VERSION 2 – REVIEW

REVIEWER	Ziggi Ivan Santini The Danish National Institute of Public Health, University of Southern Denmark, Denmark
REVIEW RETURNED	12-Feb-2021
GENERAL COMMENTS	I recommend the authors summarize their findings in the conclusion, this part is missing.

	I also recommend that they reconsider their use of the term "on the contrary". Judging from the context, it would appear they need something like "in contrast, another study found..." or "however" or something similar. I would recommend the authors drop the term "abstaining" in "abstaining from marriage", because it doesn't sound right. Rather use something like "Japan has seen a drop in marriage..." because abstaining implying a conscious (often religious) choice Apart from that, paper can be published
--	---

VERSION 2 – AUTHOR RESPONSE

Dear Reviewer 1,
Dr. Ziggi Ivan Santini, University of Southern Denmark

General response:

We appreciate your thorough consideration and constructive feedback on our manuscript. We highlighted all changes to the revised manuscript in yellow.

Comment #1:

I recommend the authors summarize their findings in the conclusion, this part is missing.

Response #1:

Thank you for your advice. We added a summary of the findings to the conclusion.

(pp 28, lines 334 to 341) We examined the association between social isolation and depression onset among older adults in England and Japan, who experience different cultural contexts regarding social relationships, and found a significant association in both countries; we also observed that in England, poor interaction with children was marginally associated, and in Japan, poor interaction and lack of social participation were significantly associated with depression. Tackling social isolation must be prioritised to safeguard the mental health of older adults worldwide. Particularly in Japan, the promotion of interaction with children and social participation could be key factors in addressing social isolation.

Comment #2:

I also recommend that they reconsider their use of the term "on the contrary". Judging from the context, it would appear they need something like "in contrast, another study found..." or "however" or something similar.

Response #2:

Thank you for your comment. We revised the content that included "on the contrary" in the manuscript. Additionally, our manuscript was checked by a native speaker throughout.

(pp 8, lines 83 to 85) In contrast, a study of older adults in the United States (US) and Japan demonstrated that while relationships with children were associated with a low level of depression only in Japan, ...

(pp 24, lines 271 to 273) However, a previous study reported no association between the presence of children and depression among older adults in the US.[22]

(pp 25 to 26, lines 291 to 293) However, the protective effects of social participation on mental health vary depending on the type of organisation with which an individual is involved,[48]...

Comment #3:

I would recommend the authors drop the term "abstaining" in "abstaining from marriage", because it doesn't sound right. Rather use something like "Japan has seen a drop in marriage...." because abstaining implying a conscious (often religious) choice

Response #3:

Thank you for your useful suggestions. We agree with you and have revised as follows.

(pp 7, lines 62 to 66) In contrast, Japan, now a super-aged society (more than 21% of the population aged 65 or above),[13] is experiencing a rapidly increasing trend in the number of never-married persons and weakening community and neighbourhood relations,[14] leading to a rise in the number of socially isolated individuals.[15]

We would like to thank the reviewers for their helpful comments and hope that the revised manuscript is acceptable for publication in BMJ Open.